

# CMTCN: a web tool for investigating cancer-specific microRNA and transcription factor co-regulatory networks

Ruijiang Li[*], Hebing Chen[*], Shuai Jiang, Wanying Li, Hao Li, Zhuo Zhang, Hao Hong, Xin Huang, Chenghui Zhao, Yiming Lu and Xiaochen Bo

Department of Biotechnology, Beijing Institute of Radiation Medicine, Beijing, China
[*] These authors contributed equally to this work.

## ABSTRACT

Transcription factors (TFs) and microRNAs (miRNAs) are well-characterized trans-acting essential players in gene expression regulation. Growing evidence indicates that TFs and miRNAs can work cooperatively, and their dysregulation has been associated with many diseases including cancer. A unified picture of regulatory interactions of these regulators and their joint target genes would shed light on cancer studies. Although online resources developed to support probing of TF-gene and miRNA-gene interactions are available, online applications for miRNA-TF co-regulatory analysis, especially with a focus on cancers, are lacking. In light of this, we developed a web tool, namely CMTCN (freely available at http://www.cbportal.org/CMTCN), which constructs miRNA-TF co-regulatory networks and conducts comprehensive analyses within the context of particular cancer types. With its user-friendly provision of topological and functional analyses, CMTCN promises to be a reliable and indispensable web tool for biomedical studies.

## INTRODUCTION

Gene expression regulation is a complex biological process involving various regulators across multiple levels. Because it controls organism development and cell homeostasis (*Yu et al., 2008*), gene expression dysregulation is closely associated with disease processes. In gene expression regulation system, transcription factors (TFs) and microRNAs (miRNAs) have been recognized to play important roles at transcriptional level and post-transcriptional level respectively. Moreover, increasing evidence suggests that miRNAs and TFs are able to work together, mainly to buffer gene expression and/or adjust signaling (*Bracken, Scott & Goodall, 2016*). Specifically, miRNAs and TFs have been shown to regulate shared target genes in feed-forward-loops (FFLs) and co-regulating pairs (*Zhang et al., 2013*). At the network level, miRNA-TF FFLs and co-regulating pairs are major network motifs (i.e., genetic interconnection patterns that occur more often by chance in biological networks), serving as basic building blocks of a complex regulatory system

Corresponding authors
Yiming Lu, ylu.phd@gmail.com
Xiaochen Bo, boxc@bmi.ac.cn

(*Anastasiadou, Jacob & Slack, 2017*; *Guo et al., 2016*). Hence, perturbations of the interwoven regulatory patterns involving miRNAs and TFs trigger global alterations in gene expression and are associated with many diseases, including cancer.

Cancer is a complex, heterogeneous disease whose etiology involves diverse genetic and environmental factors. In the complex cancer-related gene expression regulation networks, miRNAs and TFs can work cooperatively as oncogenes or tumor suppressors (*Yan et al., 2012*). The construction and analysis of miRNA-TF co-regulatory networks may be used to improve our understanding of tumorigenesis and may suggest novel therapeutic targets. Indeed, analyses of FFLs and co-regulatory patterns have already revealed an essential role of their combined regulatory influence in some well-studied cancers. For example, in colorectal cancer, *Wang et al. (2017)* found that aberrant expression of two miRNAs (hsa-mir-25 and hsa-mir-31), one TF (BRCA1), and two other genes (ADAMTSL3 and AXIN1) affected patient survival, and thus provided clues regarding the components that determine colorectal cancer prognosis. Additionally, employing FFL detection and glioblastoma multiforme-specific co-regulatory network construction and analysis, *Sun et al. (2012)* discovered that the miRNA hsa-mir-34a plays a key role in glioblastoma multiforme, a lethal form of primary brain cancer. Likewise, employing miRNA-TF co-regulatory network analysis in breast cancer, Qin et al. found novel potential breast cancer driver genes (*Qin, Ma & Chen, 2015*).

Several web resources have been developed to unravel how miRNAs and TFs interact with genes, including resources for TF-gene regulation (*Bovolenta, Acencio & Lemke, 2012*; *Han et al., 2017*; *Jiang et al., 2007*; *Zheng et al., 2008*), and numerous tools for obtaining miRNA targets by experiments and predictions (*Chou et al., 2017*; *Jiang et al., 2009*; *Xiao et al., 2009*; *Yang et al., 2011*). Although the identification of TF and miRNA targets is a key step in studying miRNA-TF co-regulation, there remains a need to combine these two forms of basic regulatory information together technically to enable identification of co-regulatory relationships and establish co-regulatory networks. Although combining co-regulatory information with disease-related knowledgebases is critical for biomedical research, online tools based on these ideas are lacking.

Here, we report the design, development, and testing of an online application called CMTCN. CMTCN collects and integrates the published regulatory relationships among miRNAs, TFs, and target genes from 11 databases and provides a means of curating cancer-specific interactions by referring to documented cancer-related gene and miRNA databases. It conducts systematic explorations of major co-regulatory motifs, namely co-regulating pairs and FFLs that consist of miRNAs, TFs, and cancer-related genes. By identifying co-regulatory interactions, CMTCN can establish miRNA-TF co-regulatory networks for cancers and provide useful analyses for understanding the molecular mechanisms underlying cancer pathogenesis.

## MATERIAL AND METHODS

### Design and workflow

CMTCN was developed by way of a five-step computational pipeline (Fig. 1). In step one, CMTCN utilized information provided by established regulatory databases of both

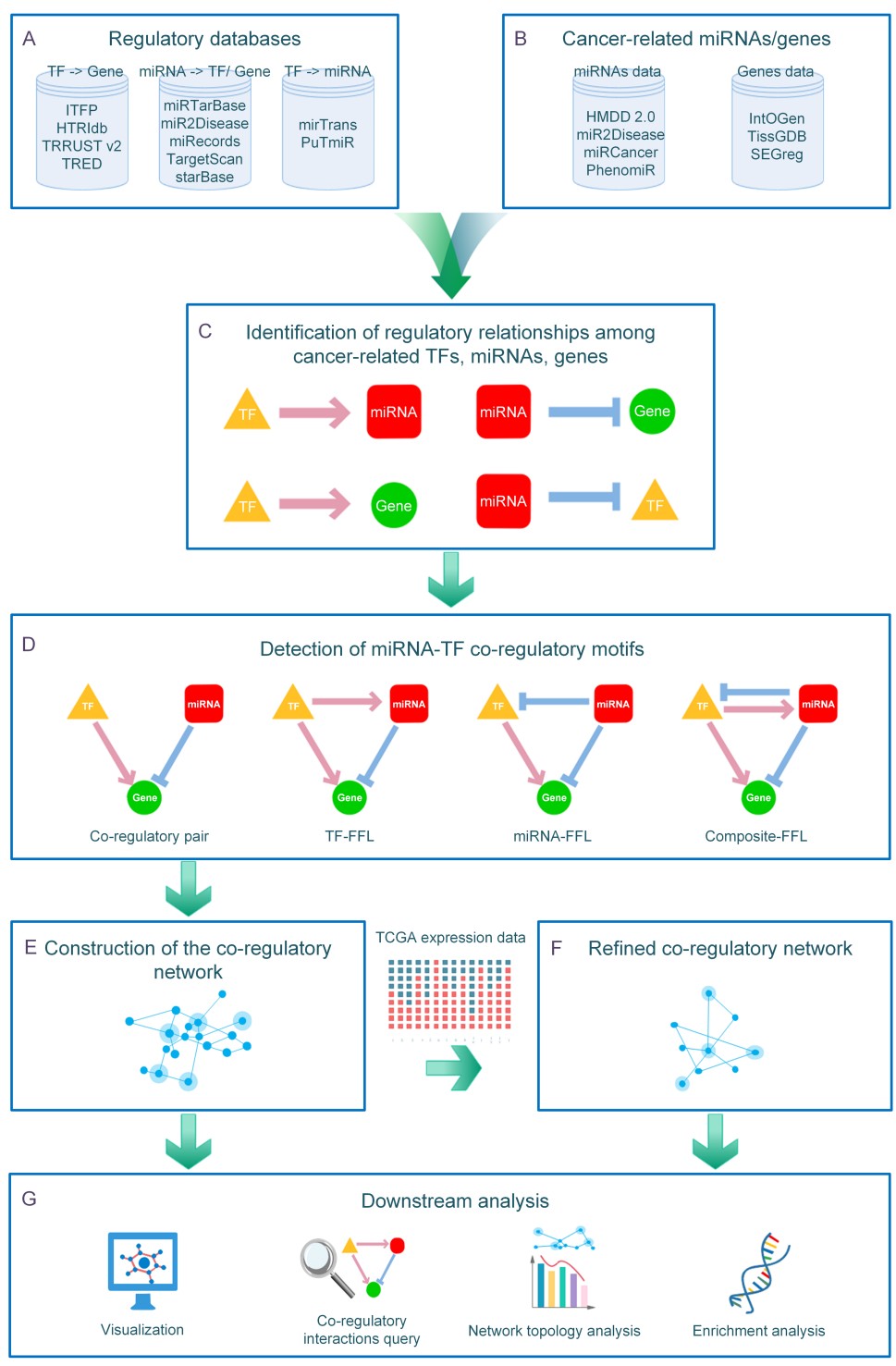

**Figure 1** **Overview of the CMTCN workflow.** (A) CMTCN utilized information provided by established regulatory databases of both predicted and experimentally validated interactions. (B) CMTCN curated cancer-related genes and miRNAs for 33 types of cancers by referring to established cancer genes/miRNAs databases. (C) CMTCN screened out cancer-related regulatory interactions (continued on next page...)

**Figure 1 (…continued)**
whose target nodes or regulator nodes are known to be relevant to cancer, forming an entirely synthetic network by pooling four types of interactions. (D) CMTCN identified FFLs and co-regulatory pairs from the combinatorial network using a network motif detection algorithm. (E) By identifying co-regulatory interactions, CMTCN can establish miRNA-TF co-regulatory networks in different cancers. (F) CMTCN incorporated expression data from TCGA to refine discoveries. (G) CMTCN supports enriched network-centric downstream analysis, including cancer-specific co-regulatory network displays, network topology analyses, co-regulatory interactions queries, and intra–co-regulatory network gene/miRNA enrichment analyses.

predicted and experimentally validated interactions (Fig. 1A, Table S1). In total, 67,770 TF-gene, 177,724 TF-miRNA, 630,106 miRNA-gene, and 97,580 miRNA-TF interactions were collected. In step two, CMTCN curated cancer-related genes/miRNAs manually for 33 types of cancer by referring to cancer gene/miRNA databases, including TissGDB (*Kim et al., 2017*), SEGreg (*Tang et al., 2018*), IntOGen (*Gonzalez-Perez et al., 2013*), HMDD v2.0 (*Li et al., 2014*), miR2Disease (*Jiang et al., 2009*), PhenomiR (*Ruepp et al., 2010*), and miRCancer (*Xie et al., 2013*) (Fig. 1B, Tables S2–S3). In step three, CMTCN screened out cancer-related regulatory interactions whose target nodes or regulator nodes are known to be relevant to cancer, forming an entirely synthetic network by pooling four types of interactions (Fig. 1C). Finally, in step four, CMTCN identified FFLs and co-regulatory pairs from the combinatorial network using the network motif detection algorithm FANMOD (*Wernicke & Rasche, 2006*) and, in step five, constructed the co-regulatory network and incorporated expression data from The Cancer Genome Atlas (TCGA) (*Katarzyna, Patrycja & Maciej, 2015*) to refine its discoveries (Figs. 1D–1F).

The online CMTCN interface is a neat and user-friendly dashboard layout with two main modules: Start and Analysis. Users initiate their research in the 'Start' module with a three-step job submission process. After the job has been submitted, the webserver jumps to the 'Analysis' module where there is access to network-centric analysis, including a cancer-specific co-regulatory network display, network topology analysis, a co-regulatory interactions query, and enrichment analysis of genes and miRNAs in a co-regulatory network (Fig. 1G).

## Data input

The user initiates a cancer-specific miRNA-TF co-regulation analysis through the construction of a co-regulatory network. CMTCN displays the co-regulatory network and provides detailed investigation for the network (Fig. 2).

The user begins in the 'Start' module with the following three steps: (i) choose a specific cancer; (ii) select a regulatory data source; and (iii) choose whether to analyze the full cancer-specific co-regulatory miRNA-TF network or to view the co-regulatory network for specific genes/miRNAs of interest (Fig. 2A). Currently, CMTCN supports 33 types of cancer. For co-regulatory network construction, users are given the option of three evidence levels (validated, predicted, or both) and two angles (full co-regulation network or co-regulatory subgraph). When the full network is selected, CMTCN provides an overall co-regulatory network for a specific cancer. When the co-regulatory subgraph is selected, users can view the genes/miRNAs they are interested in. To facilitate analyses, CMTCN

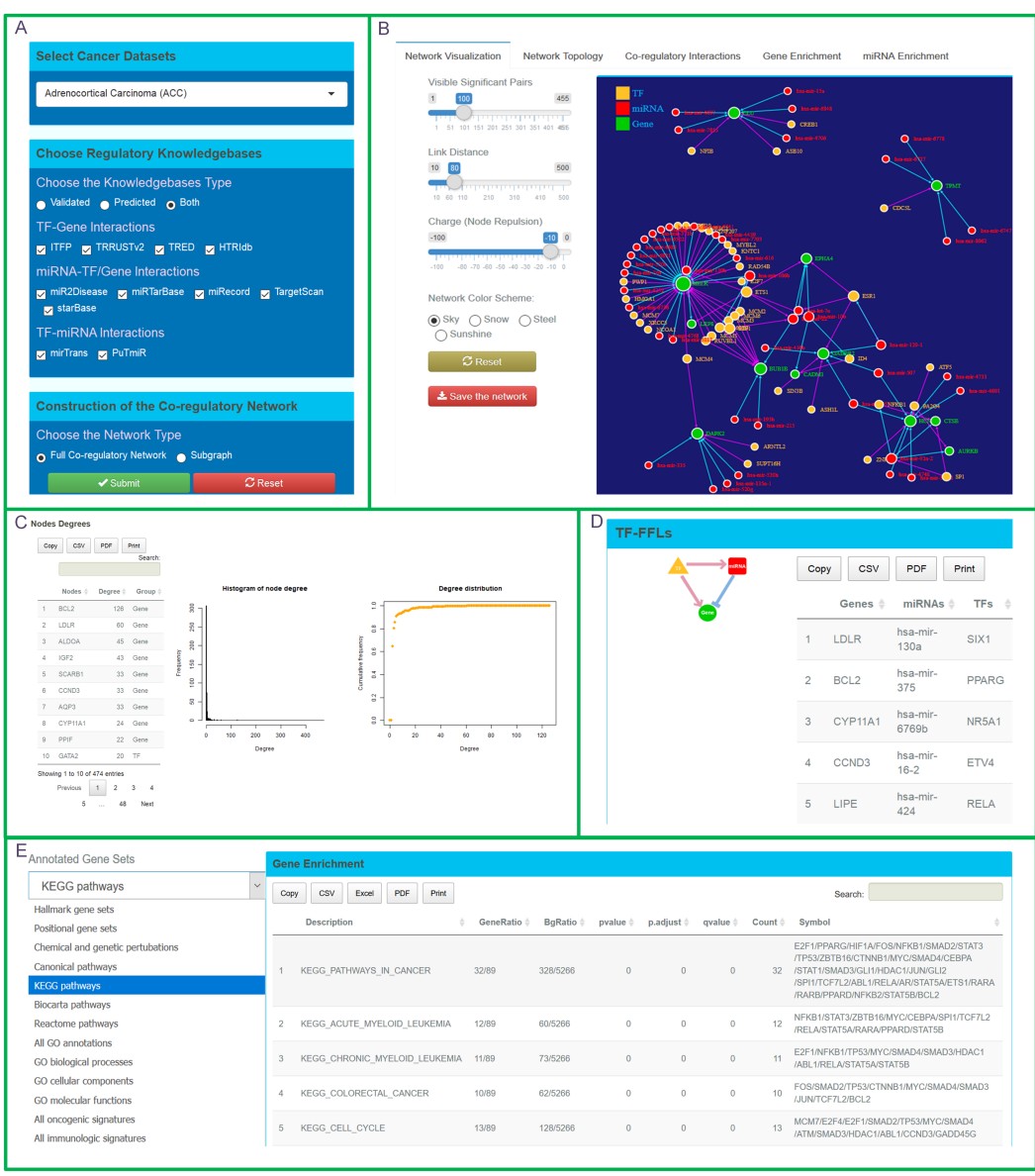

**Figure 2 Features of the interactive CMTCN web service.** (A) Users initiate their study by three steps. First, the user selects an CMTCN-supported cancer type (currently, 33 to choose from), selects the desired evidence levels, and selects whether they want to study an entire co-regulation network or a subnet of co-regulatory network for genes of interest. (B) CMTCN displays an interactive and intuitive force network map for the co-regulatory network. (C) CMTCN uses three indicators to analyze the key nodes of the established co-regulatory network. (D) CMTCN can query each co-regulatory interaction type. (E) CMTCN makes functional enrichment analysis for genes, TFs, and miRNAs involved in the co-regulatory network.

provides the gene sets associated with clinical stage (*Lee et al., 2015*) and high mutation rates (*Sun, Li & Wang, 2018*) for each type of cancer. Users can input the genes they are interested in directly or upload a gene list in the form of a .txt file for query. The sample .txt file is provided in Data S1.

## Functionality of CMTCN

### Identification of miRNA-TF co-regulatory interactions

CMTCN pools TF-gene, TF-miRNA, miRNA-gene, miRNA-TF regulatory relationships. Based on these relationships, CMTCN focuses on co-regulatory pairs and three-node FFLs identified with the help of the network motif detection algorithm. A co-regulatory pair includes a TF and miRNA that regulate a gene simultaneously. There are three types of three-node FFLs: TF-FFLs, miRNA-FFLs, and composite-FFLs. In a TF-FFL, the TF is the master regulator, which regulates a partner miRNA and their joint target. In a miRNA-FFL, the miRNA is the master regulator, repressing its partner TF and their joint target gene. A TF-FFL and miRNA-FFL can combine to form a composite-FFL in which the miRNA and TF regulate each other. CMTCN can query and display each co-regulatory pattern in detail, and can incorporate expression data from TCGA to refine co-regulatory interactions (Fig. 2D).

### Co-regulatory interactions refinement

CMTCN capitalizes on TCGA expression data to select important co-regulatory interactions. TCGA RNA-Seq data (run date 2016-01-28) provided in the Firehose data repository are accessed using the R package RTCGAToolbox (Samur, 2014). CMTCN calculates pairwise Spearman correlation values between TFs, miRNAs, and genes. Users can refine co-regulatory pairs or FFLs on the basis of correlation $p$-values and correlation coefficients. For instance, when the user sets the $p$-value cutoff to 0.05 and the correlation coefficient cutoff to 0.2, CMTCN displays a TF-target edge of $p < 0.05$ with correlation coefficients whose absolute values are $\geq 0.2$. Since most miRNAs are assumed to inhibit the expression of their targets (Beermann et al., 2016), CMTCN shows miRNA-target edge $p < 0.05$ and correlation coefficient $\leq -0.2$. Users can indicate which types of TF-target regulation they need. CMTCN gives the user the ability to select to differentiate between positive and negative TF regulation. Thus, if the user needs to examine only positive or negative regulation, CMTCN can retain only positive or negative correlation coefficient interactions, respectively.

### Network visualization

CMTCN utilizes major co-regulatory motifs to form a cancer-specific miRNA-TF co-regulatory network. It uses the D3.js to depict an interactive and intuitive co-regulatory network map in which genes, TFs, and miRNAs are represented by green, yellow, and red nodes, respectively. To improve the presentation of the force-directed graph, users can adjust link distance, node repulsion, and the number of co-regulatory relationships displayed. CMTCN network graphs presented can be saved as images (Fig. 2B).

### Network topology analysis

The key nodes in a co-regulation network have biological significance because they are signal convergence sites with pronounced control and influence over the network; accordingly, they represent potential candidates for biomarker prediction, clinical prognosis, and treatment (Barabási, Gulbahce & Loscalzo, 2011). CMTCN uses three indicators in its key node analysis: node degree, hub score, and authority score (Kleinberg, 1999) (Fig. 2C).

Node degree represents the number of edges that meet at a vertex. A node with a high hub score contains a large number of outgoing links, and a node with a high authority score is pointed to by many other nodes with high hub scores. Letting A be the adjacency matrix of the graph, the hub score is defined as the principal eigenvector of $AA^T$, and the authority score is the principal eigenvector of $A^TA$. CMTCN uses appropriate pictures to produce a vivid presentation of scoring results.

### Gene/miRNA enrichment analysis

To better capture and mine biological roles of a co-regulatory network, CMTCN takes advantage of annotated gene/miRNA sets from GSEA (*Subramanian et al., 2005*) and miEAA (*Backes et al., 2016*), thereby enabling functional enrichment analysis of genes, TFs, and miRNAs involved in the co-regulatory network. CMTCN enables detailed gene-ontology association analyses with a variety of biological and biomedical ontologies, extending beyond GO (*Consortium et al., 2000*) and KEGG (*Kanehisa & Goto, 2000*), thereby providing clues for follow-up studies (Fig. 2E).

## Implementation

The CMTCN website can be accessed freely and readily by all users without a login requirement. It supports the most prevalent web browsers, including Google Chrome, Mozilla Firefox, Safari, and Internet Explorer (10 or later). It adjusts automatically to the layout of particular browsers and device types, from desktop computers to tablets and smart phones. CMTCN was written almost entirely in R code based on the R-Shiny web framework (*Chang et al., 2017*) and has been deployed on an Aliyun server. The backend database is implemented with SQLite (version 3.8.8.2).

## RESULTS

### Functional use case of CMTCN

To better illustrate the functionality and utility of CMTCN, we studied the miRNA-TF co-regulation of two specific cancers, namely thyroid carcinoma (THCA) and ovarian cancer (OV).

### Uncovering and analyzing the miRNA-TF co-regulatory network in THCA

THCA is a common endocrine malignancy with an increasing worldwide incidence (*Cabanillas, Mcfadden & Durante, 2016*). In CMTCN, we chose the THCA cancer set, selected the validated regulation information confidence level, and built a full miRNA-TF co-regulatory network for THCA. For each type of co-regulatory pattern, we required a $p$-value $<0.05$, an absolute value of correlation coefficient $\geq 0.2$, and both types of TF regulation. CMTCN established a THCA-specific miRNA-TF co-regulatory network comprised of 391 nodes and 518 links, with 710 co-regulatory pairs, 7 TF-FFLs, 1 miRNA-FFL, and 2 composite-FFLs.

CMTCN then used network topology analysis to reveal the top-five genes in terms of authority score (MELK, PIGR, SNX5, CLU, and DAPK2) (Table S4). A comprehensive literature review of these genes confirmed their implicated roles in cancer diagnosis and therapy. MELK has been reported to be potential therapeutic targets for malignancies (*Pitner*
*et al., 2017*). PIGR has the potential to be a candidate prognostic biomarker (*Fristedt et al., 2014*). Regulation of CLU by oncogenes and epigenetic factors has important consequences for mammalian tumorigenesis (*Sala et al., 2009*). The aberrant methylation, and hence silencing, of DAPK2 has been reported to play a critical role in thyroid cancer tumorigenesis and progression (*Hu et al., 2006*). Finally, reduced expression of SNX5 was shown recently to be related to promotion of thyroid tumorigenesis (*Jitsukawa et al., 2017*) and SNX5 expression studies can be used to support a pathology diagnosis of thyroid cancer (*Ara et al., 2012*). Additionally, CMTCN carried out a functional enrichment analysis for genes and TFs in the THCA-specific miRNA-TF co-regulatory network. With KEGG pathway enrichment, CMTCN found 11 significant pathways, all of which were related to cancer (Table S5). CMTCN pinpointed four TFs (E2F4, TFDP1, SP1, MYC) and one gene ACVR1 in the transforming growth factor-β signaling pathway, a negative regulator of thyroid follicular cell growth (*Geraldo, Yamashita & Kimura, 2012*).

### MiRNA-TF co-regulatory subnetwork of top mutated genes in OV

OV is highly aggressive gynecological cancer (*Sung et al., 2017*). We used CMTCN to establish an OV-specific miRNA-TF subnetwork encompassing the top-100 mutated genes in OV. Again, we set the confidence level to validated and required a *p*-value <0.05, an absolute correlation coefficient ≥0.2 and both types of TF regulation. Our goal was to use CMTCN to reveal the miRNAs and TFs related to the top mutated genes in OV, as well as the regulatory effects of these miRNAs and TFs.

We obtained six co-regulated pairs and one TF-FFL related to the top-100 mutated genes in OV, which revealed six miRNAs and three TFs with possible associations with these top mutated genes. The sole TF-FFL obtained was comprised of a TF (TP53), a miRNA (hsa-mir-29c), and a joint target gene (PTEN). In this TF-FFL, the TF regulates both the miRNA and the target gene, with the miRNA repressing the target gene. Regarding OV pathogenesis, the loss function of PTEN, together with *TP53* alteration is a common event (*Martins et al., 2014*). Interestingly, hsa-mir-29c, an effector of regulator TP53, can also suppress cancer development (*Li et al., 2018*). The possibility that abnormal expression of the two cross-talking regulators and their co-target gene may be predictive of OV risk is worth further careful study in future experiments.

## DISCUSSION

The results of these demonstration studies, described before, show that CMTCN is able to uncover and analyze miRNA-TF co-regulation networks in a manner that can enhance our understanding of miRNA-TF gene regulatory mechanisms in different types of cancer and provide valuable information for cancer prognosis and therapy.

CMTCN explored miRNA-TF co-regulatory pairs and FFLs systematically and in a context-specific manner. To enhance the power and accuracy of the discovery, CMTCN provides TCGA expression-based filtering options for calculations of pairwise correlations between miRNAs, TFs, and genes. Owing to its simplicity and large-scale network computing capability, like other related analyzation methods (*Qin, Ma & Chen, 2015*; *Wang et al., 2017*), CMTCN uses pairwise correlations to refine co-regulation. In addition

to the refinement, CMTCN combines network topology information with co-regulatory relationship queries to provide a sum of degree, hub, and authority scores for each co-regulation interaction type, which supports the discovery of high-value co-regulatory interactions. In fact, there are multiple ways to deal with co-regulatory interaction mining outcomes and there are opportunities to improve the co-regulatory analysis framework in future work. Methods, such as partial correlation or the emerging detrended partial-cross-correlation analysis (DPCCA) method (*Yuan et al., 2015*), could be applied in the refinement step. Moreover, integrating our miRNA-TF co-regulatory network with other functional networks will potentiate the findings at a systems level.

## CONCLUSIONS

Here, we introduced CMTCN as a user-friendly online tool for miRNA-TF co-regulation analysis in the context of cancer research. CMTCN characterized and detected co-regulatory pairs and three types of FFLs for each type of cancer. It constructed detailed and dynamic cancer-specific miRNA-TF co-regulatory networks that elucidate the interwoven pivotal roles of TFs, genes, and miRNAs in human cancer. CMTCN identified pivotal network nodes and prioritized those nodes that should be investigated further experimentally as potential biomarkers or drug targets. The program supports various enrichment analyses for discovery of network gene/miRNA ontology associations. Though it was developed for miRNA-TF co-regulation analysis studies specifically, CMTCN has broad biomedical applications and can be utilized by cancer researchers as well as systems biologists and epigenetic scholars. Cancer researchers can utilize CMTCN to find candidate cancer genes; systems biologists can explore the qualities of the comprehensive network-centric analyses of CMTCN. Epigeneticists can use CMTCN to interpret the integrative global effects of TFs and miRNAs on cancer.

### Funding
This work was funded by the General Program (31771397, 81573251) of the Natural Science Foundation of China (http://www.nsfc.gov.cn), the Major Research plan of The National Natural Science Foundation of China (No. U1435222), the Program of International S&T Cooperation (No. 2014DFB30020) and the National High Technology Research and Development Program of China (No.2015AA020108). The funders had no role in study design, data collection and analysis, decision to publish, or preparation of the manuscript.

### Grant Disclosures
The following grant information was disclosed by the authors:
Natural Science Foundation of China: 31771397, 81573251.
Major Research plan of The National Natural Science Foundation of China: U1435222.
Program of International S&T Cooperation: 2014DFB30020.
National High Technology Research and Development Program of China: 2015AA020108.

## Competing Interests

The authors declare there are no competing interests.

## Author Contributions

- Ruijiang Li conceived and designed the experiments, performed the experiments, contributed reagents/materials/analysis tools, prepared figures and/or tables, authored or reviewed drafts of the paper, approved the final draft.
- Hebing Chen conceived and designed the experiments, authored or reviewed drafts of the paper.
- Shuai Jiang performed the experiments.
- Wanying Li, Hao Li and Zhuo Zhang analyzed the data.
- Hao Hong, Xin Huang and Chenghui Zhao contributed reagents/materials/analysis tools.
- Yiming Lu conceived and designed the experiments, authored or reviewed drafts of the paper.
- Xiaochen Bo conceived and designed the experiments, authored or reviewed drafts of the paper, approved the final draft.

## Data Availability

GitHub: https://github.com/rilletlee/CMTCN.

## Supplemental Information

Supplemental information for this article can be found online at http://dx.doi.org/10.7717/peerj.5951#supplemental-information.

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
