# Peer review of "CMTCN: a web tool for investigating cancer-specific microRNA and transcription factor co-regulatory networks"

_PeerJ, doi:10.7717/peerj.5951_

## Round 0.1 · original submission · Minor Revisions

We would be glad to consider a revised version of the manuscript in light of the minor comments by both the reviewers.

Reviewer 1 ·

Basic reporting

The resources gathered by the authors is highly valuable. The authors should make available to download in a human and/or computer readable format (delimited text file such as csv, tsv or SIF and perhaps, though less important, an XML based format such as SBML):
1. the currently analyzed network
2. the set of resources used

Experimental design

Overall, methods are insufficiently described.

It is unclear how the network associated with a specific disease is built. In particular, the refinement step is not described. One can suppose it is done by removing any mir->gene or tf->gene edge that has a Pearson correlation of 20% or less or a correlation p-value above 5%, but this needs to be clearly stated.

Are positive correlation between miRNA and genes or TFs simply discarded?

For usability purposes, the authors limit by default the number of visible co-regulatory pairs. How are these ordered to only show 100 by default?

Identifying genes associated with drug-response is not a trivial task. The tool proposes an input list containing such genes. How were these gene sets defined? To what drug are they associated?

Validity of the findings

How are TF dealt with when these are among the input gene list? Are they considered only as TF? Could it be possible to add co-regulatory pairs and FFL with TFs as target genes?

Gene enrichment and network motifs are valuable resources. However, listing hundreds of motifs can make the analysis difficult. It would be interesting to prioritize TF and miRNAs based on the number of co-regulatory pairs it appears in, as well as to order the mirna TF pairs by the number of motifs in which they both appear in. One of the values of CMTCN is the possibility to see how TF-miRNA co-regulatory interactions are disrupted in a particular disease. Therefore, it would greatly help to be able to analyze the most relevant pairs first. As for now, scrolling through a table with hundreds to thousands of entries is too tedious for this type of analysis.

The knowledgebases column in the co-regulatory interactions table should be split in two, one for TF and one for miRNA DB.

It is important to differentiate positive and negative regulation of TF. For instance, TF-FFL with a positive or negative regulation of a miRNA has broadly different biological implications.

Additional comments

The authors propose a novel aggregated resources for TF and miRNA gene regulation as well as an efficient online application to analyze cancer-specific networks. The app is well designed, easy to use and fairly reactive. The analysis of the co-regulation between TF and miRNA is an interesting approach and the proposed app greatly facilitates its analysis in a context-specific manner.

·

Basic reporting

The manuscript is well written and uses clear unambiguous language. There are a few cases were it seems that a wrong word was used, e.g.

Line 26: “the disorder of gene expression regulation systems has concerned closely with disease” should probably be something like “the disorder of gene expression regulation systems is closely associated with diseases.”

Line 146: “prevailing web browsers” - probably means “prevalent web browsers”.

Experimental design

It should be made clear which version of each database is currently used. I suggest adding a separate tab where the sources are listed with source URL and date of download.

Similarly, it is not clear from where the TCGA data was obtained (source, date of download). It should also be made clear how these data were processed. This information should be added to both the web application as well as the manuscript.

In the methods section (line 120) it is not explained how TCGA data is actually used to refine co-regulatory interactions. Only when using the application does it become clear that correlation thresholds (and corresponding p-values) are used.

CMTCN considers only pairwise correlation between TFs, miRNAs and genes. However, in triplets partial correlation could be used to isolate the contribution of a specific regulator. This should be briefly discussed.

In the methods section, it should be explained how exactly survival-relevant, drug response related and top mutated genes have been identified.

Validity of the findings

No comment

Additional comments

With CMTCN, Li et al present a shiny-based web application that allows its users to study TF-miRNA co-regulatory pairs and feed-forward-loops for 33 cancer types. The web application is well designed and intuitive. It is well documented and provides all functionality I would expect from such an application. During my testing I did not find any issues or bugs. The web application is certainly a useful feature for cancer researchers.

The only issue I have is that the web application does not clearly indicate when it is processing data in the background. Such background data processing could be indicated to the user with an overlay, for instance, also to avoid that the users just continues clicking on other things.

I strongly recommend the authors make the code for their web application available via github or a similar code repository.

---

## Round 0.2 · accepted · Accept

All the issues raised by the reviewers have been successfully addressed and the manuscript can be now endorsed for publication.

Reviewer 1 ·

Basic reporting

no comment

Experimental design

no comment

Validity of the findings

no comment

Additional comments

The authors answered to all the comments that were raised.

·

Basic reporting

no comment

Experimental design

no comment

Validity of the findings

no comment

Additional comments

The authors have addressed all of my previous comments.